# Alterations in Plasma Lipidomic Profiles in Adult Patients with Schizophrenia and Major Depressive Disorder

**DOI:** 10.3390/medicina58111509

**Published:** 2022-10-24

**Authors:** Fei Wang, Lin Guo, Ting Zhang, Zhiquan Cui, Jinke Wang, Chi Zhang, Fen Xue, Cuihong Zhou, Baojuan Li, Qingrong Tan, Zhengwu Peng

**Affiliations:** 1Department of Psychiatry, Chang’an Hospital, Xi’an 710000, China; 2Department of Psychiatry, Xijing Hospital, Air Force Medical University, Xi’an 710032, China; 3School of Biomedical Engineering, Air Force Medical University, Xi’an 710032, China

**Keywords:** lipidomics, plasma lipid, schizophrenia, major depressive disorder

## Abstract

*Background and Objectives:* Lipidomics is a pivotal tool for investigating the pathogenesis of mental disorders. However, studies qualitatively and quantitatively analyzing peripheral lipids in adult patients with schizophrenia (SCZ) and major depressive disorder (MDD) are limited. Moreover, there are no studies comparing the lipid profiles in these patient populations. *Materials and Method*: Lipidomic data for plasma samples from sex- and age-matched patients with SCZ or MDD and healthy controls (HC) were obtained and analyzed by liquid chromatography-mass spectrometry (LC-MS). *Results*: We observed changes in lipid composition in patients with MDD and SCZ, with more significant alterations in those with SCZ. In addition, a potential diagnostic panel comprising 103 lipid species and another diagnostic panel comprising 111 lipid species could distinguish SCZ from HC (AUC = 0.953) or SCZ from MDD (AUC = 0.920) were identified, respectively. *Conclusions*: This study provides an increased understanding of dysfunctional lipid composition in the plasma of adult patients with SCZ or MDD, which may lay the foundation for identifying novel clinical diagnostic methods for these disorders.

## 1. Introduction

Schizophrenia (SCZ) and major depressive disorder (MDD) are the leading cause of morbidity of mental disease worldwide [1,2]. SCZ affects approximately 1% of the world’s population, whereas the lifetime risk of MDD is 15–18% [3,4]. Despite there being many studies on the pathogenesis of SCZ and MDD, the diagnosis for these disorders relies mainly on the subjective interpretation of clinical symptoms presented by patients. Although SCZ and MDD are two separate diagnostic entities that can be defined by their clinical features, converging evidence suggests that these disorders have overlapping characteristics in symptom presentation and neurocognitive impairments [5,6]. For example, motivational and hedonic impairments are present in individuals with MDD and SCZ [7,8], and their severity is associated with the development of both psychosis and depression [9,10]. In contrast, depressive symptoms are common in SCZ, and psychotic symptoms are also part of the clinical presentation observed in MDD [11]. Previous studies reported that the prevalence of depressive symptoms was up to 20–60% in patients with SCZ [12,13] and may occur in all disease courses [14]. Accordingly, the prevalence of psychotic features in adolescent MDD is 18%, and the lifetime prevalence of psychotic depression varies between 0.35% and 1% [15]. Therefore, investigating the objective biological differences between these disorders may help to develop new diagnostic methods.

In addition to clinical symptoms, using plasma as a sample source to identify diagnostic for SCZ and MDD has already been reported [16,17,18]. Lipid plays an important role in neuronal development and brain function [19,20,21]. Moreover, plasma lipids were already represented as a target for the treatment of depression or psychotic symptoms in SCZ and MDD [22], and lipidomics has been established as a pivotal tool for diagnosing SCZ and MDD [23,24]. However, there are only a few indirect understandings of the potential role of plasma lipids in the pathophysiology and no studies have directly compared the lipid compositions of adult patients with SCZ and MDD. Such investigations might be valuable for understanding the shared and objective peripheral biomarkers for these two disorders.

We performed a case-control study that enrolled age-matched adult patients with SCZ (*n* = 31), MDD (*n* = 35), and healthy controls (HC, *n* = 32), and analyzed the plasma lipidomics by LC-MS. We aimed to thereby determine the differences in lipid composition and investigate the correlation between differential lipids and clinical symptoms. Moreover, we aimed to identify SCZ and MDD- elated lipidomic signatures by using orthogonal partial least squares discrimination analysis (OPLS-DA). Finally, we aimed to identify discriminative lipid panels that could distinguish individuals with SCZ, those with MDD, and HCs using random forest and receiver operating characteristic (ROC) analysis.

## 2. Materials and Methods

### 2.1. Subjects and Sampling

This study was registered, and its protocol was approved by Chinese Clinical Trial Registry (ChiCTR2000032118) and Ethics Committee (ChiECRCT20200090), respectively. It was performed in accordance with the tenets of the Declaration of Helsinki and all subjects volunteered to take part and provided written informed consent. The inclusion criteria for the SCZ were: (1) compliance with the diagnostic and statistical manual of mental disorders (DSM)-5 diagnostic standards for schizophrenia or schizophreniform disorder; (2) the positive and negative symptoms scale (PANSS) score was greater than or equal to 60; (3) age of 18 to 65 years; and (4) no history of probiotics, probiotic fermented food, or any antibiotics within 1 month. The inclusion criteria for MDD were: (1) compliance with the DSM-5 diagnosis standards for major depressive disorder; (2) the Hamilton Depression Rating Scale (HAM-D) score was greater than or equal to 18; (3) age of 18 to 65 years; and (4) no history of probiotics, probiotic fermented food, or any antibiotics within 1 month. The exclusion criteria were (1) obesity, body mass index (BMI) ≥ 28.0; (2) high-fat diet partisans and vegetarians; (3) hypertension; (4) alcohol abuse or dependence; (5) illicit drug use; (6) menstruation, pregnancy or lactation; and (7) presence of other mental disorders. Moreover, the structured clinical interview for DSM-5, Hamilton Anxiety Scale (HAM-A), HAM-D, and PANSS was independently administered by two psychiatrists who were blinded to the clinical trial grouping. The same exclusion criteria were applied to HCs.

Finally, 31 SCZ (12 male and 19 female, age 22–45 years) and 35 MDD (age 12 male and 23 female, 25–57 years), along with 32 HCs (10 male and 22 female, age 22–53 years), were recruited from the Department of Psychiatry in Chang’an Hospital. All of whom underwent a physical examination. The preexisting psychiatric disorders were screened by Mini-International Neuropsychiatric Interview. Participants were prohibited from eating and drinking after 10 PM and the blood samples were collected between 8 AM and 10 AM next day. The concentration of cholesterol (CHOL), triglycerides (TG), low-density lipoprotein (LDL), and high-density lipoprotein (HDL) were detected immediately by blood lipid routine examination. Moreover, blood was collected in anticoagulant tubes and centrifuged (1600 rpm, 15 min). The obtained plasma was stored in liquid nitrogen tank until lipidomics analysis and the temperature at which the plasma was stored was less than −190 °C.

### 2.2. Lipidomics Analysis

The experiments were performed as described previously [25,26] and the data analysis was supported by Majorbio Bio-pharm and Shanghai Applied Protein Technology Co., Ltd. In brief, plasma (100 μL, accurately measured) was spiked with internal lipid standards (SPLASH^®^ LIPIDOMIX^®^ Mass Spec Standard, methanol solution, AVANTI, 330707-1EA, Merck, Darmstadt, Germany) and then homogenized with appropriate amounts of water, methanol and methyl tert-butyl ether for sample preparation. Then, samples were submitted for ultrasonication and centrifuged (14,000× *g*, at 10 °C for 15 min), and the supernatant was separated for the LC-MS analysis. Samples were separated by using Nexera LC-30A system with a C18 column (ACQUITY UPLC CSH C18, 130Å, 1.7 μm, 2.1 mm × 100 mm, Waters), column temperature 45 °C with a flow rate of 300 μL/min. The lipid extracts were re-dissolved in 200 μL of 90% isopropanol/acetonitrile, centrifuged for 15 min (14,000× *g*), and finally, 3 μL of the sample was injected. To avoid the influence of the instrument detection signal fluctuation, a random sequence was used to continuously analyze the samples. Q-Exactive Plus (Thermo Fisher Scientific, San Jose, CA, USA) was used to acquire mass spectra and electrospray ionization (ESI) parameters were optimized and preset for all measurements. Meanwhile, single-point internal standard calibrations were used to estimate the absolute concentrations of unique lipids identified by accurate mass spectrometry, MS/MS spectral matching, and retention times [27], and lipid identification was performed using LipidSearch^TM^ software (Thermo Fisher Scientific, San Jose, CA, USA).

### 2.3. Statistical Analyses

Statistical analyses were performed using SPSS 21.0 software (IBM-SPSS Inc, Chicago, IL, USA) (Kruskal-Wallis test for nonnormal distribution and one-way analysis of variance for normal distribution). Lipid data were further converted using log10 and standardized using Pareto scaling. Supervised partial least squares discriminant analysis (PLS-DA) was first used to demonstrate the overall distribution between samples and the stability of the entire analytical process. A supervised OPLS-DA was then used to identify differential lipids between groups. To prevent the model from overfitting, the OPLS-DA models were validated by permutation analysis (200 times). The identified lipids with variable importance for the projection (VIP) scores of > 1.5 in the OPLS-DA, and values of FDR (two-tailed Student’s *t*-test) < 0.05 and fold change (FC) > 2 or <0.5 were regarded as differentially abundant lipids. To obtain simplified potential biomarker panels, the identified differentially abundant lipids were used to conduct stepwise logistic regression analysis. The diagnostic performance of these identified panels was conducted to assess the receiver operating characteristic (ROC) curve analysis, and R package was used for AUC calculation and illustration (R-3.5.3). The Pearson correlation analysis was used to assess the correlations between the clinical parameters and differentially abundant lipids. The online software MetaboAnalyst 5.0 (https://www.metaboanalyst.ca/ (accessed on 26 January 2022)) was used to conduct KEGG pathway analysis and pathway enrichment analysis.

## 3. Results

### 3.1. Clinical Characteristics of the Recruited Participants

Ninety-eight individuals were included in the study. There were no significant differences in terms of age (*p* = 0.066), sex (*p* = 0.803) and BMI (*p* = 0.573) among the three groups. Meanwhile, there were also no significant differences among the three groups in terms of TG (*p* = 0.114), HDL (*p* = 0.079) and CHOL (*p* = 0.632). PANSS, HAM-D, and HAM-A scores in the SCZ and MDD groups were higher than those in the HC group, and LDL in MDD was also higher than that in the HC group (Table 1). The main therapeutic drugs used in the MDD group were duloxetine and venlafaxine, while that in the SCZ group was risperidone. There is no significant difference between SCZ and MDD at the duration of illness, which was defined as the time since the first occurrence of symptoms. Notably, none of the recruited participants smoked, and there was no significant difference among the groups in terms of smoking or exposure to secondhand smoke in the past 6 months (*p* = 0.130). The detailed clinical and demographic characteristics of the study cohort are shown in Appendix A.

### 3.2. Alternation of Lipid at Class Level in SCZ and MDD

Thirty lipid classes and 782 lipid species were identified (Appendix A). Lipidomic analysis revealed that there were significant differences in the levels of eight lipid classes, including acylcarnitine (AcCa, *H* = 31.696, *p* < 0.001) (Figure 1A), ceramide (Cer, *H* = 12.257, *P* = 0.002) (Figure 1C), phosphatidylethanolamine (PE, *H* = 15.268, *p* < 0.001), lysophosphatidylethanolamine (LPE, *H* = 29.497, *p* < 0.001), lysophosphatidylcholine (LPC, *H* = 13.021, *p* = 0.001), lysophosphatidylinositol (LPI, *H* = 11.0861, *p* = 0.002), phosphatidylinositol phosphate (PIP, *H* = 19.937, *p* < 0.001), and phosphatidylinositol 4,5-bisphosphate (PIP2, *H* = 9.123, *p* = 0.010) (Figure 1E). Intercomparison between SCZ and HC groups further showed that the levels of AcCa and PE were low, whereas those of LPC, LPE, LPI, PIP, and PIP2 were higher in the SCZ when compared with that in HC. Similarly, levels of AcCa and PE were decreased, whereas those of PIP and PIP2 were increased in the MDD group when compared with HC. Notably, the levels of Cer, LPC, LPE, and LPI were increased in the SCZ group when compared with the MDD group.

Furthermore, levels of LPC, LPE, and PIP were positively correlated while the levels of PE and AcCa were negatively correlated with the PANSS score (Figure 1F). Meanwhile, the concentrations of PS and monogalactosyldiacylglycerol (MGDG) were negatively correlated with positive scores of PANSS, whereas PIP2 levels were positively correlated. Moreover, the concentrations of PS were negatively correlated, whereas Cer levels were positively correlated with negative scores of PANSS. In the MDD group, concentrations of PE and AcCa were negatively correlated with both HAM-A and HAM-D scores, whereas concentrations of PIP and GM3 were positively correlated. Additionally, the concentrations of PIP2 were positively correlated with the HAM-D score (Appendix A). Taken together, the levels of eight lipids in the SCZ group and four lipids in the MDD group were changed compared to the HC group at the class level. Moreover, the concentrations of seven lipids in the SCZ group were changed compared to those in the MDD group.

### 3.3. Characteristic Lipid Species between SCZ and HC

The plasma lipid data could be distinguished between the SCZ and HC groups according to the characteristics of the OPLS-DA model (Figure 2A). Moreover, the permutation test intercept R^2^ = 0.458 and Q^2^ = −0.684, which can better reflect the robustness of the model. A total of 103 differential lipid species were identified between the SCZ and HC (45 upregulated and 58 downregulated in SCZ, Figure 2B and Appendix A). Correlation analysis showed the relevance of those discriminated lipids and clinical scale scores (Appendix A). Enrichment analysis of the Kyoto Encyclopedia of Genes and Genomes (KEGG) pathway based on these differential lipids indicated that glycerophospholipid metabolism, linoleic acid metabolism, alpha-linolenic acid metabolism, and glycosylphosphatidylinositol (GPI)-anchor biosynthesis were enriched in the SCZ group (Figure 2C). Meanwhile, the above molecules were selected to perform ROC curve analysis and showed good sensitivity and specificity (AUC = 0.953) (Figure 2D), indicating that these 103 plasma lipids might be a combinational biomarker for SCZ.

### 3.4. Characteristic Lipid Species in MDD and HC Groups

Plasma lipid data were also distinguished and reliable for screening lipid biomarkers between the MDD and HC groups (Figure 3A). Meanwhile, the permutation test intercept R^2^ = 0.842 and Q^2^ = −0.851, which can better reflect the robustness of the model indicating the OPLS-DA model was also reliable for screening lipid biomarkers in the MDD and HC groups. Four different lipid species (three upregulated and one downregulated in MDD) were identified between the MDD and HC groups (Figure 3B and Appendix A) and there was no significant enrichment of lipid metabolism between these two groups. Correlation analysis showed the relevance of the above four lipids and clinical scale scores (Appendix A). However, these four lipids showed limited sensitivity and specificity (AUC = 0.622, 95% CI (0.480, 0.765)) (Figure 3C) and could not be used as a potential combinational plasma biomarker for MDD.

### 3.5. Characteristic Lipids between SCZ and MDD Groups

Plasma lipid data were well distinguished and reliable for screening lipid biomarkers in patients with SCZ and MDD (Figure 4A). In total, 111 lipid species were identified (58 upregulated and 53 downregulated in MDD) between the SCZ and MDD groups (Figure 4B and Appendix A). Correlation analysis showed the relevance of those lipids and clinical scale scores (Appendix A). Enrichment analysis of the KEGG pathway based on these differential lipids indicated that glycosylphosphatidylinositol (GPI)-anchor biosynthesis, alpha-linolenic acid metabolism, linoleic acid metabolism, and glycerophospholipid metabolism were enriched in the SCZ group (Figure 4C). Meanwhile, the above-identified lipid species were selected to construct the conduct ROC curve and showed good specificity and sensitivity (AUC = 0.920) (Figure 4D), indicating that these plasma lipids might be useful parameters to distinguish MDD and SCZ.

## 4. Discussion

In the present study, we investigated the plasma lipid composition of adult patients with SCZ and MDD compared to each other and compared with HCs using quantitative validation and comprehensive lipid profiling based on LC-MS. Moreover, we identified potential diagnostic plasma lipids, which can distinguish SCZ patients from MDD patients and SCZ patients from HCs with high reliability. However, only four lipids have been identified and it could not distinguish MDD patients from HC, suggesting that lipidomic analysis may be one of the useful methods to identify SCZ. The results are worth further exploration with a large sample size in clinical research as that may help to further identify potential diagnostic molecular targets for SCZ and MDD.

With the continuous development of COVID-19, the prevalence of mental diseases continues to increase, and it has become an important problem that seriously affects social stability and economic development. However, the differentiation and diagnosis of mental diseases still mainly depend on clinical manifestations and doctors’ clinical diagnosis and treatment level. Therefore, the investigation of objective molecular markers will provide assistance for the differential diagnosis of mental diseases. Recently, alternations in lipid composition in patients with mental diseases, such as bipolar disorder, SCZ, and MDD have been reported [28,29,30]. A direct comparison of the lipid compositions in patients with SCZ and MDD is particularly valuable for understanding the common and unique lipid characteristics of these two diseases and identifying lipid markers that might differentiate SCZ, MDD, and HC subjects. In the present study, we firstly found that levels of AcCa and PE were decreased in the SCZ and MDD groups compared with the HC group and were negatively correlated with PANSS, HAM-D, and HAM-A scores. Supporting these findings, previous studies found that levels of PE were decreased in the serum and postmortem prefrontal white matter in SCZ patients [31,32], and serum PE was also negatively correlated with depressive symptoms [33]. Meanwhile, previous studies have shown reduced AcCa in the serum or plasma of patients with MDD than in HCs [34,35], and changes in AcCa are involved in the action of antidepressants in MDD patients [36]. Likewise, AcCa also plays a key role in the pathophysiology of SCZ [37], and AcCa supplementation therapies have been used in individuals with SCZ [38]. PE is the second most abundant phospholipid in mammalian membranes and represents the backbone of most biological membranes. For example, mitochondria have a higher PE content than other organelles [39]. Previous studies found that AcCa plays an essential role in transporting long-chain fatty acids across the mitochondrial inner membrane during β-oxidation [40] whereas mitochondrial PE induces changes in mitochondrial morphology in mammalian cells [41]. Therefore, decreases in AcCa and PE may be involved in mitochondrial dysfunction in patients with SCZ and MDD [42]. Furthermore, we also found that the levels of PIP and PIP2 were increased in both SCZ and MDD patients compared to HCs. Although there is no direct evidence indicating the involvement of PIP and PIP2 in the pathogenesis of SCZ and MDD, previous studies have found that PIP and PIP2 play fundamental roles in cell biology, including membrane-delineated signal transduction, as well as regulation of membrane trafficking and cytoskeletal dynamics [43,44]. These results indicate that there are several similar changes in lipid classes in SCZ and MDD, and these lipids primarily belong to membrane lipids and are related to oxidative stress, energy metabolism, and neurodegenerative diseases.

Importantly, the change in lipid class between SCZ and HC groups was more evident than that between MDD and HC groups. Interestingly, the levels of LPC, LPE, and LPI were decreased in both the HC and MDD groups compared with the SCZ group, which were positively correlated with the PANSS score. LPC, LPE, and LPI are prominent components of lysophospholipids that are involved in membrane function, apoptosis, oxidative stress, and inflammatory responses. Previous studies have found that LPC and LPE may protect neurons against ischemic-induced oxidative stress [45]. However, recent studies have also found that LPC polarizes macrophage activation toward the M1 phenotype, and induces inflammatory cytokine expression [46,47], which may lead to atherosclerosis and cardiovascular diseases [48]. Moreover, LPC mediates vascular barrier disruption and demyelination [49,50], which may lead to SCZ. Meanwhile, high serum levels of LPE also significantly enhance the rate of mild cognitive impairment in Alzheimer’s disease [51]. Likewise, LPI is an endogenous ligand for the G protein-coupled receptor 55, which has anti-inflammatory effects in cultured microglia but induces inflammatory cytokines in macrophages [52,53]. Therefore, the functions of lysophospholipids are inconsistent or even contradictory in different diseases or locations. Contrary to the results of the present study, a previous study found that plasma LPC and LPE levels increased after olanzapine treatment in female antipsychotic-naïve first-episode patients with SCZ [54]. Previous studies have indicated increased plasma Cer levels in MDD [55] and Cer might be one of the potential antidepressant targets both in the peripheral and brain [56,57]. Unfortunately, the present study did not observe an increase in Cer levels in MDD when compared with HC groups, and the levels of Cer were even lower in the MDD group than in the SCZ group. Nevertheless, the results of lipidomics in SCZ have been inconsistent. Previous studies found that fatty acid catabolism was upregulated in the serum of schizophrenia patients and plasma TG and LPC were increased in young adults with psychotic experiences [58,59]. However, another lipidomic study showed no significant difference between twin pairs discordant for SCZ and healthy twins in the abundance of PE, SM, LPC, and TG in serum samples [60]. Yet another study found that there was no significant difference in the concentrations of PC, PE, TG, and FA between drug-naïve patients with the first episode of SCZ and HCs and between patients with chronic SCZ who did not adhere to prescribed medications and HCs [61]. These discrepancies may be related to differences in the methods used for lipidomic analysis and the age, sex, or dietary structure of the subjects enrolled. Of course, discrepancies owing to whether the patients were first-episode patients and/or the interference of drugs cannot be ruled out.

In terms of lipid species, the plasma lipid data can be divided into three groups (Appendix A). In accordance with the trend of changes in lipid classes, only four differential lipid species were identified between the MDD and HC groups, whereas 103 species were identified between the SCZ and HC groups. In addition, 111 differential lipid species were also identified in patients with MDD and SCZ. The comparison of lipids in all three groups showed four unique lipids in MDD patients versus HCs, ten unique lipids in SCZ patients versus HCs, and eighteen unique lipids in MDD versus SCZ patients (Appendix A). A previous study identified 37 differentially regulated lipids for drug-free MDD patients in serum lipidomic analysis, and another study identified 18 differential lipids for drug-free SCZ patients in plasma lipidomic analysis [23,62]. Therefore, here, we report some previously unreported lipid signatures associated with patients with SCZ or MDD. This study lays a possibility for further characterization of the shared and distinct plasma lipid underpinnings of SCZ and MDD. In addition, we wanted to know what lipid biomarkers could discriminate between SCZ and MDD patients (from each other and HCs) in order to develop a potential diagnostic tool. Toward this end, we identified a signature of 103 lipids that could distinguish SCZ from HC (AUC = 0.953) as well as a signature of 111 lipids that could distinguish SCZ from MDD (AUC = 0.920). We believe that these signatures have the potential for distinguishing SCZ from MDD patients (and SCZ from HCs), which would be helpful to fill the clinical need to quickly distinguish SCZ from MDD to optimize the initial treatment approach. However, these identified lipid panels were inconsistent with a previous study [63], which may be related to the sex, region, and symptom characteristics of enrolled individuals as well as the analysis method, which need to be further verified by clinical studies with larger samples in the future.

Treatments aimed at modulating the composition and function of lipids might be useful to prevent or improve individuals from developing SCZ and MDD [64,65]. However, the crosstalk between the peripheral and central lipid homeostasis remains unclear and lipidomic analysis of the cerebrospinal fluid might provide further insight. Furthermore, age and sex are key factors that influence lipid metabolism [66,67]. Although there was no significant difference in age and sex among individuals with MDD, individuals with SCZ and HCs included in this study, the characteristics of lipid composition under other age conditions and sex composition are not clear. Furthermore, other signatures that can discriminate individuals with MDD and SCZ from HCs and each other, such as gut microbiota and tryptophan metabolism, have been recently reported [68,69,70], and the interaction between plasma lipid metabolism and these signatures needs to be further explored.

Nevertheless, accumulated evidence reported the influence of antidepressants and antipsychotics on lipid metabolism [71,72,73], and antidepressants appear to have fewer unfavorable effects than second-generation antipsychotics on lipid metabolism [74]. In the present study, risperidone was the main therapeutic drug used in the SCZ group and its effects on lipid metabolism have been reported [75]. Venlafaxine and duloxetine were the main therapeutic drugs used in the MDD group. Although their effects on lipid metabolism were poorly investigated, their potential impact on lipidomics in MDD cannot be excluded. Thus, it is necessary to exclude the influence of drugs on lipidomics by including the patients with the first untreated MDD or SCZ and the time effect and dose effect of the above drugs on lipid metabolism also need to be investigated as well. On the other hand, there is no significant difference in terms of TG, HDL, and CHOL among HC, MDD, and SCZ. However, levels of LDL in MDD were higher than that in the HC group and the influence of LDL on lipidomics between these two groups needs to be explored in the future.

In addition, several limitations of this study should be mentioned. Firstly, the number of recruited participants is relatively small, and the difference in plasma lipidome between subtypes of MDD and SCZ could not be analyzed. Meanwhile, our findings did not reveal the causal relationship between the difference in lipid compositions in SCZ or MDD patients and disease development, a limitation that is inherent to any cross-sectional study of this nature. Furthermore, we did not evaluate the symptoms of depression and anxiety in SCZ by HAM-D and HAM-A. Although PANSS general psychopathology contains the items of “anxiety” and “depression”, it is difficult to compare the symptom severity accessed by HAM-D scores and PANSS. Therefore, analysis of the correlation between levels of changed lipids and anxiety and depression is limited. Nevertheless, non-targeted lipidomics and lipid identification by LipidSearch will bring some speculative results, a well-known current technical problem, which leads to discrepancies in the molecules identified in different studies. For example, SM (d14:0/22:0) was elevated in the blood of the nonalcoholic steatohepatitis group compared to those in the nonalcoholic fatty liver group [76]. Meanwhile, CerG2GNAc1 (d32:1) was increased while CerG2GNAc1 (d41:4) was decreased in the plasma of patients with Fabry disease compared with the control [77]. These molecules rarely appeared in other studies, suggesting that the results of lipidomics only have reference significance, and its clinical application will only be possible with future progress in quantitative technology. Finally, lipidomics examination based on plasma LC-MS requires expensive equipment and professional operators, and the analysis of the results takes nearly 3 days. Therefore, although plasma lipidomics might be a potential tool in differentiating depression and schizophrenia, its clinical application is still difficult at this stage.

## 5. Conclusions

We identified the different plasma lipid compositions in adult patients with SCZ versus MDD and in patients with SCZ versus HCs. Moreover, we developed lipid classifiers that can effectively discriminate patients with SCZ from MDD and HCs.

## Figures and Tables

**Figure 1 medicina-58-01509-f001:**
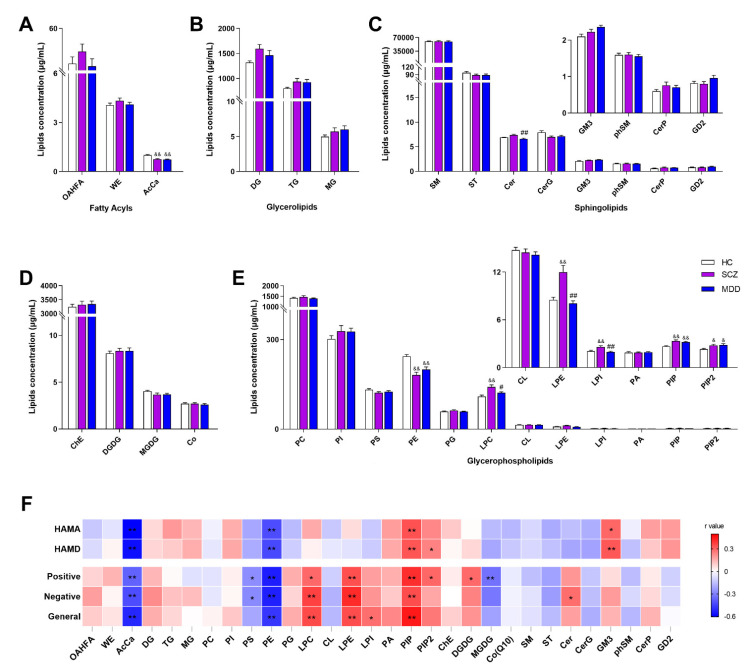
The comparison of lipid class among SCZ, MDD, and HC participants. (**A**) Fatty acyls, (**B**) glycerolipids, (**C**) sphingolipids, (**D**) ChE, Co, DGDG, and MGDG, (**E**) glycerophospholipids, and (**F**) correlation between clinical parameters and levels of lipid classes. AcCa, acylcarnitine; Cer, ceramides; CerG, glucosylceramides; CerP, ceramide phosphate; ChE, cholesterol ester; Co, coenzyme; CL, cardiolipin; DG, diglyceride; DGDG, digalactosyldiacylglycerol; LPC, lysophosphatidylcholine; LPE, lysophosphatidylethanolamine; LPI, lysophosphatidylinositol; MG, monoglyceride; MGDG, monogalactosyldiacylglycerol; OAHFA, (O-acyl)-1-hydroxy fatty acid; PA, phosphatidic acid; PC, phosphatidylcholine; PE, phosphatidylethanolamine; PG, phosphatidylglycerol; PI, phosphatidylinositol; PIP, phosphatidylinositol; PIP2, phosphatidylinositol 4,5-bisphosphate; PS, phosphatidylserine; phSM, phytosphingomyelin; SM, sphingomyelin; ST, sulfatide; TG, triglyceride; WE, wax esters. HC, healthy control; SCZ, schizophrenia; MDD, major depressive disorder; ^&^ *p* < 0.05 vs. HC, ^&&^ *p* < 0.01 vs. HC, ^#^ *p* < 0.01 vs. SCZ, ^##^ *p* < 0.01 vs. SCZ. Blue squares show a negative correlation and red squares show a positive correlation, * *p* < 0.05; ** *p* < 0.01.

**Figure 2 medicina-58-01509-f002:**
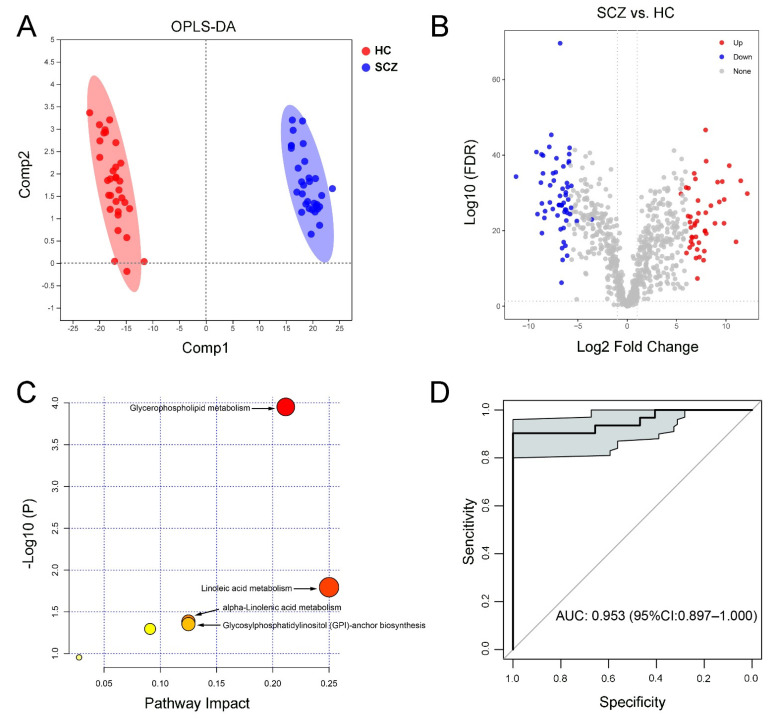
Differential lipid species and their functions between SCZ and HC. (**A**) OPLS-DA model for SCZ and HC, (**B**) volcano map showed the differential species (red dots indicated the increased lipid species and blue dots indicated the decreased lipid species) between SCZ and HC, (**C**) KEGG pathway enrichment analysis, and (**D**) ROC analysis for the combinational lipids.

**Figure 3 medicina-58-01509-f003:**
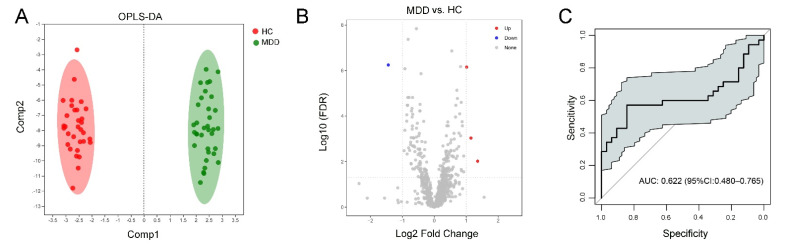
Differential lipid species between MDD and HC. (**A**) OPLS-DA model for MDD and HC, (**B**) volcano map showed the differential species (red dots indicated the increased lipid species and blue dots indicated the decreased lipid species) between MDD and HC participants, and (**C**) ROC analysis for the combinational lipids.

**Figure 4 medicina-58-01509-f004:**
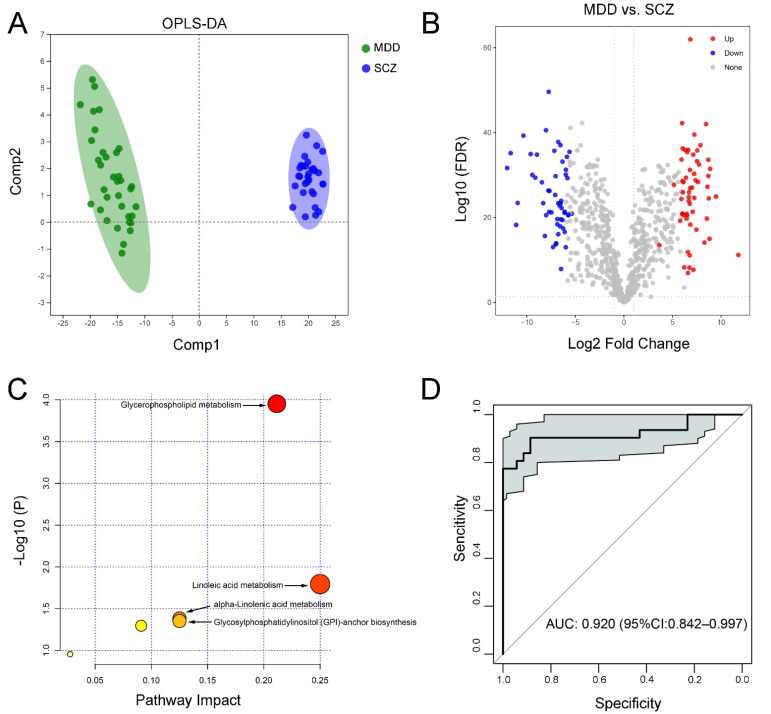
Differential lipid species and their functions between MDD and SCZ participants. (**A**) OPLS-DA model for MDD and SCZ, (**B**) volcano map showed differential species (red dots indicated the increased lipid species and blue dots indicated the decreased lipid species) between MDD and SCZ, (**C**) KEGG pathway enrichment analysis, and (**D**) ROC analysis for the combinational lipids.

**Table 1 medicina-58-01509-t001:** The comparison of clinical characteristics and symptom scale assessment among HC, MDD, and SCZ.

Parameter	HC (*n* = 32)	MDD (*n* = 35)	SCZ (*n* = 31)	H/χ^2^/F/Z Value	*p*-Value
Sociodemographic					
Age [years, M (*P*_25_, *P*_75_)] ^a^	29 (25, 36.75)	34.5 (28.5, 38)	28 (24, 37)	H = 5.435	0.066
Gender (male/female) ^b^	10/22	12 / 23	12 / 19	χ^2^ = 0.391	0.803
BMI [kg/m^2^, [M (*P*_25_, *P*_75_)]] ^a^	20.7 (18.89, 23.55)	21.45 (18.75, 23.23)	20.76 (19.28, 24.46)	H = 1.113	0.573
Marital status (single/married) ^b^	13/19	2/33	22/9	χ^2^ = 29.953	<0.001
Duration of illness [mouths, M (*P*_25_, *P*_75_)] ^c^	-	36 (12, 60)	48 (24, 48)	Z = −0.466	0.641
Smoking or exposure to secondhand smoke situation in the past six months ^b^	24/8	19/16	23/8	χ^2^ = 4.228	0.13
TG [mmol/L, M (*P*_25_, *P*_75_)] ^a^	0.91 (0.7, 1.27)	0.82 (0.7, 1.29)	1.21 (0.69, 2.01)	H = 4.35	0.114
LDL ^c^	2.21 ± 0.91	2.89 ± 0.90 *	2.50 ± 0.66	F = 5.802	0.004
HDL ^a^	1.25 ± 0.38	1.41 ± 0.28	1.28 ± 0.21	H = 5.071	0.079
CHOL ^c^	4.18 ± 0.84	4.32 ± 0.84	4.12 ± 0.86	F = 0.461	0.632
Scale evaluation					
HAM-D [M (*P*_25_, *P*_75_)] ^d^	3 (2, 5)	25 (21, 26)	-	Z = −6.926	<0.001
HAM-A [M (*P*_25_, *P*_75_)] ^d^	4.5 (2.25, 6.75)	25 (20, 29)	-	Z = −7.045	<0.001
PANSS total score [mean ± SD (range)] ^a^	35.78 ± 3.63 (30–42)	59.11 ± 10.84 (37–80) *	72.74 ± 17.99 (38–121) *	H = 64.436	<0.001
PANSS positive scale [M (*P*_25_, *P*_75_)] ^a^	7 (7, 9)	7 (7, 7)	20 (16, 25) *^#^	H = 68.09	<0.001
PANSS negative scale [M (*P*_25_, *P*_75_)] ^a^	8 (7, 9)	11 (7, 14) *	15 (12, 20) *^#^	H = 42.988	<0.001
PANSS general scale [M (*P*_25_, *P*_75_)] ^a^	19.5 (17.25, 22)	42 (35, 46) *	37 (34, 43) *	H = 61.577	<0.001

Abbreviations: ^a^ Kruskal–Wallis; ^b^ chi-square tests; ^c^ one-way ANOVA; ^d^ Mann–Whitney U; * *p* < 0.05 vs. HC group; ^#^ *p* < 0.05 vs. MDD group; BMI, body mass index; SD, standard deviation; CHOL, cholesterol; TG, triglycerides; LDL, low-density lipoprotein; HDL, high-density lipoprotein; values are shown as mean ± SD or M (*P*_25_, *P*_75_).

## Data Availability

Not applicable.

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
