# Peer review of "Alterations in Plasma Lipidomic Profiles in Adult Patients with Schizophrenia and Major Depressive Disorder"

_medicina, 2022, doi:10.3390/medicina58111509_

Round 1

Reviewer 1 Report

This is a vigorous paper based on their detailed knowledge and information about libidemics, analyzed samples about 100 people they collected.

I want to suggest a few minor points that should be corrected or added.

Line 27.

Schizophrenia (SCZ) and major depressive disorder (MDD) are the leading causes of morbidity worldwide [1, 2].”

You should change "morbidity" to "morbidity of mental desease".

Lines 77-78

"(2) a severely imbalanced diet"

It should be clearly described how the invalanced diet was evaluated. Did you use any established and standardized assessment scale?

Line 85

"35 MDD (age 12 male 85 and 19 female, 25-57 years),"

The number of MDD groups does not match; rewrite to fit with Table 1.

Line 90.

"under fasting conditions"

This statement needs more information. For example, what time were they prohibited from eating and drinking the day before blood collection?

Lines 93-94

"The obtained plasma was stored in liquid nitrogen until lipidomics analysis."

Please be more specific about the temperature at which the plasma was stored.

Below what degrees centigrade?

Line 117.

"abnormally distributed"

'Nonnormal distribution' or 'skewed distribution' are more common representations.

Table 1

1. The headings "Sociodemographic" and "Scale evaluation" are very difficult to read. Please open a line or draw a line or put them to the left.

2. Many items (the mean age and P25 for HC, P75 for MDD, P25,75 for SCZ, all items listed as Course of disease, and P25,P75 for HAM-D and HAM-A scores.

) are integers, such as 24.00, which are unnatural. If you want the significant figures to be decimal points less than 2 digits, please fill in the two digits correctly.

However, since the range of PANSS scores is supposed to be an integer, decimal points need not be listed. (For example, 20.00-42.00 is fine as 32-42.)

3. The term "course of disease" is not commonly used in psychiatric papers; it should be changed to "duration of illness".

4. HAMD and HAMA should be written as HAM-D and HAM-A, consistent with the text body.

5. This paper aims to identify biomarkers that distinguish between depression with psychotic symptoms and schizophrenia with depressive symptoms. If you want to investigate them, it is incongruous that the PANSS subitems were not written. PANSS positive symptoms, negative symptoms, and general psychopathology scores should be also written. Most of psychiatric papers usually have these information. 

6. The fact that the HAM-D was not tested in schizophrenia group is also an important limitation of this paper. PANSS general psychopathology contains the items of "anxiety" and "depression", so you may add them on the table 1. About this, please discuss among your study group member.  

Lines 167,169,171,270,296.

There are numerous spelling errors in the PANSS (The correct name is PANSS).

Discussion

You wrote many previous studies about lipidemics, and its usefulness as a candidate biomarker. However, clinical application as a biomarker requires a lot of  process. Blood collection from patients, analysis, and feedback of the results. In fact, please add how feasible they are because they are not mentioned. If possible, please include about cost, the effort of analysis, and the time it takes to analyze.

Lines 396-398.

Is this necessary?

Author Response

Thanks for your patience and positive comments. Your comments are of great significance for improving this paper. The changes in the revised manuscript were highlighted in yellow. The followings are our point-by-point reply to the comments:

Line 27. "Schizophrenia (SCZ) and major depressive disorder (MDD) are the leading causes of morbidity worldwide [1, 2].”You should change "morbidity" to "morbidity of mental desease".

Response:"morbidity" was instead by "morbidity of mental desease".

Lines 77-78 "(2) a severely imbalanced diet". It should be clearly described how the invalanced diet was evaluated. Did you use any established and standardized assessment scale?

Response:We did not use any standardized assessment scale. Actually, we exclude high-fat diet partisans and vegetarians according to the diet structure. Thus, "(2) a severely imbalanced diet" was instead by "high-fat diet partisans and vegetarians".

Line 85 "35 MDD (age 12 male and 19 female, 25-57 years)," The number of MDD groups does not match; rewrite to fit with Table 1.

Response:Sorry for this mistake. It should be described as "35 MDD (age 12 male and 23 female, 25-57 years)".

Line 90. "under fasting conditions" This statement needs more information. For example, what time were they prohibited from eating and drinking the day before blood collection?

Response:That's a very important question. In the present study, individuals were prohibited from eating and drinking after 10 PM the day before blood collection. We changed the description as "Participants were prohibited from eating and drinking after 10 PM and the blood samples were collected between 8 AM and 10 AM next day." in the revised manuscript.

Lines 93-94 "The obtained plasma was stored in liquid nitrogen until lipidomics analysis." Please be more specific about the temperature at which the plasma was stored. Below what degrees centigrade?

Response:The temperature that the plasma was stored was less than 190 degrees centigrade below zero. We changed the description as "The obtained plasma was stored in liquid nitrogen tank until lipidomics analysis and the temperature that the plasma was stored was less than -190 ℃." in the last sentence of 2.1.

Line 117. "abnormally distributed" 'Nonnormal distribution' or 'skewed distribution' are more common representations.

Response:"abnormally distributed" was instead by "nonnormal distribution".

 Table 1

  1. The headings "Sociodemographic" and "Scale evaluation" are very difficult to read. Please open a line or draw a line or put them to the left.

Response: We put the headings "Sociodemographic" and "Scale evaluation" to the left.

  1. Many items (the mean age and P25 for HC, P75 for MDD, P25,75 for SCZ, all items listed as Course of disease, and P25, P75 for HAM-D and HAM-A scores.) are integers, such as 24.00, which are unnatural. If you want the significant figures to be decimal points less than 2 digits, please fill in the two digits correctly. However, since the range of PANSS scores is supposed to be an integer, decimal points need not be listed. (For example, 20.00-42.00 is fine as 32-42.)

Response: The data exhibit in Table 1 is not clear. Zero after the decimal point was not displayed in the revised manuscript.

  1. The term "course of disease" is not commonly used in psychiatric papers; it should be changed to "duration of illness".
  2. HAMD and HAMA should be written as HAM-D and HAM-A, consistent with the text body.

Response: Thank you for your suggestion. The above errors have been revised.

  1. This paper aims to identify biomarkers that distinguish between depression with psychotic symptoms and schizophrenia with depressive symptoms. If you want to investigate them, it is incongruous that the PANSS subitems were not written. PANSS positive symptoms, negative symptoms, and general psychopathology scores should be also written. Most of psychiatric papers usually have these information.

Response: Thank you for your suggestion. The comparison of PANSS positive symptoms, negative symptoms, and general psychopathology scores was added in Table 1 in the revised manuscript.

  1. The fact that the HAM-D was not tested in schizophrenia group is also an important limitation of this paper. PANSS general psychopathology contains the items of "anxiety" and "depression", so you may add them on the table 1. About this, please discuss among your study group member.

Response: That’s really a limitation of this paper. Due to schizophrenic patients in our department rarely evaluate their anxiety and depression status, we did not evaluate the HAM-D of SCZ in this study. On the other hand, the items of "anxiety" and "depression" in PANSS can reflect the emotional state of SCZ, it is difficult to compare the symptom severity accessed by HAM-D scores and PANSS. We added the description “Furthermore, we did not evaluate the symptoms of depression and anxiety in SCZ by HAM-D and HAM-A. Although PANSS general psychopathology contains the items of "anxiety" and "depression", it is difficult to compare the symptom severity accessed by HAM-D scores and PANSS. Therefore, analysis of the correlation between levels of changed lipids and anxiety and depression is limited” in the last paragraph of Discussion.

Lines 167,169,171,270,296.

There are numerous spelling errors in the PANSS (The correct name is PANSS).

Response: Sorry for the negligence. The above spelling errors have been revised.

Discussion

You wrote many previous studies about lipidemics, and its usefulness as a candidate biomarker. However, clinical application as a biomarker requires a lot of process. Blood collection from patients, analysis, and feedback of the results. In fact, please add how feasible they are because they are not mentioned. If possible, please include about cost, the effort of analysis, and the time it takes to analyze.

Response: This is a very important question. At present, lipidomics detection is limited to laboratory examination, and its clinical application is still unfeasible. We added the description “Finally, lipidomics examination based on plasma LC-MS requires expensive equipment and professional operators, and the analysis of the results takes nearly 3 days. Therefore, although plasma lipidomics might be a potential tool in differentiating depression and schizophrenia, its clinical application is still difficult at this stage.” at the end of Discussion.

Lines 396-398. Is this necessary?

Response: The last sentence in Conclusions was deleted.

Reviewer 2 Report

The manuscript entitled “Alterations in plasma lipidomic profiles in adult patients with 2 schizophrenia and major depressive disorder” by Wang and colleagues is well-designed and focusing an interesting topic. The data presented offers extensive insight into the lipid status parameters alterations that coincide with SCZ and MDD. I find this study a reliable starting point for future analyses that will allow the defining of interconnection between those elements.   

Author Response

Thanks for your patience and positive comments.

Reviewer 3 Report

This manuscript addresses  an interesting and important topic. It deals with an underresearched area. It is very well written. The background is clearly presented in the Introduction. Mehthodsare timely and appropriate. Novel findings are well presentid in text, tables and figures. Discussion of the findings is appropriate. Limitations are clearly described by the authors.

Nevertehless I raise some critical remarks.

As the authors state in the Discussion the patients are not drug free. Medication my have influenced the results. Therefore in the Abstract it should be mentionned that the patients included in this study were medicated.

Table 1 - what is meant with "course of disease"? - Is this the duration of the present episode or the time since the first occurence of symptoms of MDD or SCZ?

Minor issues

line 50 - please errase s in "understandings"

line 55 - please add "patients  with " following  "adult"

Author Response

Thanks for your patience and positive comments. The changes in the revised manuscript were highlighted in yellow. The followings are our point-by-point reply to the comments

1、As the authors state in the Discussion the patients are not drug free. Medication my have influenced the results. Therefore in the Abstract it should be mentionned that the patients included in this study were medicated.

Response: Sorry for the negligence. We have changed the description as "In this study, lipidomics data for plasma samples from sex- and age-matched and medicated patients with SCZ or MDD and healthy controls (HC) were obtained ..."

Table 1 - what is meant with "course of disease"? - Is this the duration of the present episode or the time since the first occurence of symptoms of MDD or SCZ?

Response: Course of disease was defined as the time since the first occurrence of symptoms. Therefore, we add the sentence in line 144 "There is no significant difference between SCZ and MDD at course of disease, which was defined as the time since the first occurrence of symptoms."

Minor issues

line 50 - please errase s in "understandings"

line 55 - please add "patients  with " following  "adult"

Response: These two minor mistakes have been corrected in the revised manuscript.